# Burnout Syndrome, Stress and Study Hours in the Selection Process for Educational Teaching Staff: The Role of Resilience—An Explanatory Model

Eduardo Melguizo-Ibáñez [ID], Gabriel González-Valero [ID], Félix Zurita-Ortega [ID], José Manuel Alonso-Vargas *, Maria Rosario Salazar-Ruiz and Pilar Puertas-Molero

Department of Didactics Musical, Plastic and Corporal Expression, Faculty of Education Science, University of Granada, 18071 Granada, Spain
* Correspondence: josemalonsov@correo.ugr.es

**Abstract:** Candidates for the public teaching profession are subjected to high levels of stress, which can lead to the development of burnout syndrome during the competitive examination process. The present research reflects the objective of analysing the effect of resilience on burnout syndrome, stress and study hours in Spanish public teacher candidates. A cross-sectional, descriptive, comparative and ex post facto study was carried out on a sample of 4117 Spanish candidates (M = 31.03; S.D = 6.800). The Perceived Stress Scale was used to measure the stress variable. The Maslach Burnout Inventory was used to collect data related to burnout syndrome and the Connor-Davidson Resilience Scale was used to measure data related to resilience. The conclusions are that resilience helps to mitigate the effects generated by burnout syndrome and stress, helping to maintain a positive attitude towards the number of hours of study.

**Keywords:** burnout syndrome; stress; resilience; education





## 1. Introduction

Currently, in Spain, to become a public teacher involves a competitive examination process (Real Decreto 270/2022 2022). The "*opositores*" are the applicants for the various teaching posts in the teaching corps who are examined to obtain a teaching post (Melguizo-Ibáñez et al. 2022). The aim of the competitive examination is to demonstrate knowledge of the teaching speciality for which one is applying (Real Decreto 270/2022 2022). It consists of two exams: A practical test that assesses scientific capacity and mastery of technical knowledge, together with the written development of a topic related to the speciality for which the candidate is applying (Real Decreto 270/2022 2022), and a second test consisting of the defence of a didactic programme that refers to the curriculum of an area within the speciality for which the candidate is applying, together with the preparation and presentation of a didactic unit presented by the candidate (Real Decreto 270/2022 2022).

The selection process is not an easy path as there is a high level of uncertainty and fear of failure during the preparation process (Arias and Amate-Romera 2019). This fear of failure is generated by failing to take the required exam (Arias and Amate-Romera 2019; von der Embse et al. 2019). The increase in stress levels occurs in the following phases (Selye 1975): The first is that the subject is warned to be alert to a particular stressor (Jordan et al. 2020). If this phase of alarm, or alertness, is prolonged over a period of time in which the subject copes with the situation, it leads to the resistance phase (Jordan et al. 2020) in which the subject perceives that there is a limit to his or her ability to withstand the stressor (Luceño-Moreno et al. 2020). Subsequently, the exhaustion phase is presented (Luceño-Moreno et al. 2020). It is characterised by fatigue and it also affects the motivational–affective level of the competitor (Luceño-Moreno et al. 2020).

Willroth et al. (2020) stated that being subjected to permanent stress levels can lead to negative motivational–affective change. This change can lead to a psychological state of tiredness and exhaustion, negatively influencing exam preparation (Raudales et al. 2020).

It has been observed and demonstrated that prolonged exposure over time to levels of stress and negative thoughts can lead to a state known as Burnout Syndrome (Freudenberger 1989; Puertas-Molero et al. 2022). It is characterised by loss of motivation, emotional exhaustion, physical fatigue, and low levels of tolerance and commitment to a process or task (Luceño-Moreno et al. 2020; Puertas-Molero et al. 2022). Wang et al. (2020), Martinent et al. (2020) and Capone and Petrillo (2020) state that Burnout Syndrome occurs in three dimensions: emotional exhaustion, low self-fulfilment and depersonalisation. In studies by Melnick et al. (2020), Luceño-Moreno et al. (2020) and Madigan and Kim (2021), it is stated that continued subjection to these levels of burnout and stress can develop negative effects on the mental and physical health of individuals. This can result in symptoms such as continued fatigue, muscle fatigue and psychological disorders, which can work against the jobseeker's intended purpose.

Studies have pointed out that resilience is a fundamental element to overcome disruptive effects (Fullerton et al. 2021). This concept has been defined by Román-Mata et al. (2020) and Romero-Barquero (2020) as an intrinsic capacity to overcome stressful and adverse situations in order to achieve their goals. In academia, Fullerton et al. (2021) state that resilience plays a fundamental role in the achievement of academic goals. Similarly, Mcdermott et al. (2020) and González-Valero et al. (2022a) add that factors such as self-efficacy, planning, stress management and persistence play a key role in achieving academic success. Jordan et al. (2020) assessed levels of burnout and resilience over the course of an academic year, concluding that stress levels increased significantly when an assessment test was approaching. In contrast, resilience levels were found to positively affect the avoidance of extreme levels of stress, and higher than average levels of resilience led to better academic performance and stress management (Jordan et al. 2020; Kangas-Dick and O'Shaughnessy 2020).

The aim of this study is to analyse the effect of resilience on burnout syndrome, stress and hours of study in candidates for the Spanish public teaching profession. The following research hypotheses are proposed:

**H1.** *Stress and burnout syndrome will have a negative effect on the study hours spent preparing for the selection tests.*

**H2.** *Resilience will act positively on the study hours spent on test preparation.*

**H3.** *Negative relationships will be found between resilience and burnout syndrome.*

**H4.** *Resilience will be negatively associated with stress.*

## 2. Materials and Methods

### 2.1. Design and Participants

A cross-sectional, descriptive, comparative study was carried out on candidates for the Spanish public teaching profession. The sample was made up of 4117 participants, of which 1363 (33.1%) were male and 2754 (66.9%) were female. The ages of the participants ranged from 23 to 54 years (M = 31.03; S.D = 6.800). Two inclusion criteria were established. These constituted firstly of holding a university degree in primary education and secondly of being a candidate for the Spanish public teaching profession. Finally, data were collected from January 2022 to May 2022, with the test dates being in June 2022. Table 1 shows geographical distribution of participants.

**Table 1.** Geographical distribution of participants.

|  | *N* | % |
|---|---|---|
| **La Rioja** | 14 | 0.3% |
| **Basque Country** | 30 | 0.7% |
| **Balearic Islands** | 31 | 0.8% |
| **Navarre** | 39 | 0.9% |
| **Cantabria** | 69 | 1.7% |
| **Aragon** | 73 | 1.8% |
| **Extremadura** | 110 | 2.7% |
| **Asturias** | 113 | 2.7% |
| **Canary Islands** | 126 | 3.1% |
| **Catalonia** | 158 | 3.8% |
| **Region of Murcia** | 213 | 5.2% |
| **Castille and Leon** | 254 | 6.2% |
| **Castille La Mancha** | 359 | 8.7% |
| **Galicia** | 401 | 9.7% |
| **Valencian Community** | 576 | 14.0% |
| **Community of Madrid** | 629 | 15.3% |
| **Andalusia** | 922 | 22.4% |
| **Total** | 4117 | 100.0% |

*2.2. Instruments*

***Ad hoc* socio-demographic questionnaire**. The instrument collected the gender, age, speciality for which the candidates were applying, and study hours per day devoted to the preparation for the test.

**Perceived Stress Scale (PSS)** (Cohen et al. 1983), the version adapted to Spanish by Remor (2006) was used. The scale consists of a total of 14 items that are answered on a five-point Likert scale (0 = never, 4 = very often). Regarding the reliability analysis, Cronbach's alpha scored $\alpha = 0.879$.

**Maslach Burnout Inventory (MBI)** (Maslach and Jackson 1981), the version adapted to Spanish by Seisdedos (1997) was used. It is composed of 22 items that are answered on a seven-point Likert scale (0 = never and 6 = daily). This instrument assesses Burnout Syndrome from a three-dimensional perspective: Emotional Exhaustion, Depersonalisation and Personal Accomplishment. The reliability analysis scored very well, with EE $\alpha = 0.807$, DP $\alpha = 0.861$ and PR $\alpha = 0.735$.

**Connor-Davidson Resilience Scale (CD-RISC)** (Connor and Davidson 2003), the Spanish version by Crespo et al. (2014) was used. It consists of 25 items that are answered on a five-point Likert scale where 0 = "not true at all" and 4 = "usually true". The questionnaire assesses resilience from a pentadimensional perspective, with the following areas being found: Personal Competence ($\alpha = 0.919$); High Standards and Tenacity, Confidence in one's own instincts, Tolerance of Negative Affects and Strengthening the Effects of Stress ($\alpha = 0.866$); Acceptance of Positive Change and Secure Relationships ($\alpha = 0.834$); Control and Purpose ($\alpha = 0.734$); Spiritual Influences ($\alpha = 0.542$).

*2.3. Procedure*

A literature review was carried out to address the current problem. Numerous research studies were found which have been used to contextualise and discuss the results obtained (Jiménez-Ortiz et al. 2019; González-Valero et al. 2021; Ye et al. 2021; Özhan 2021; Ruiz-Calzado and Llorent 2018; Janatolmakan et al. 2021; Román-Mata et al.

2020; González-Valero et al. 2022b; Rodríguez-Fernández et al. 2018; Vicente de Vera-García and Gabari-Gambarte 2019; León-Hernández et al. 2019; Yang and Wang 2022; Suárez-Martel and Martín-Santana 2019; Chakradhar et al. 2022; Baumgartner and Schneider 2021; Wang et al. 2022; Giorgi et al. 2020; Smith and Emerson 2021; Puertas-Molero et al. 2022; Tong et al. 2021; Valdivieso-León et al. 2020). Subsequently, the research team created a Google Form including the objective of the study and whether the participants decided to participate in the research on a voluntary basis, guaranteeing their anonymity at all times. Once this had been done, the questionnaire was sent out via the different social networks. This research study complied with the ethical principles for human research established in the Helsinki Declaration and was carried out under the supervision of the Research Ethics Committee of the University of Granada (2966/CEIH/2022).

*2.4. Data Analysis*

IBM SPSS Statics 25.0 was used for the statistical analysis. The normality was studied using the test of Kolmogorov-Smirnov, which showed a normal distribution. A descriptive analysis of the data was carried out using frequencies, followed by a relational analysis, using Pearson's bivariate test. A one-factor ANOVA was performed. Statistically significant differences were determined through Pearson's Chi-Square test ($p \leq 0.05$). The magnitude of the difference in effect size (ES) was obtained with Cohen's standardized d-index (Cohen 1992).

The IBM SPSS Amos 26.0 programme was used to establish the theoretical model (Figure 1). The model is composed of a total of five endogenous variables and two exogenous variables. For the latter type of variables, a causal explanation was carried out by focusing on the observed associations between the indicators and the degree of measurement reliability, allowing for the inclusion of the measurement error of the observed variables. The direction of the arrows symbolises the direction of the effect of the variables. The significance level was also set at 0.05.

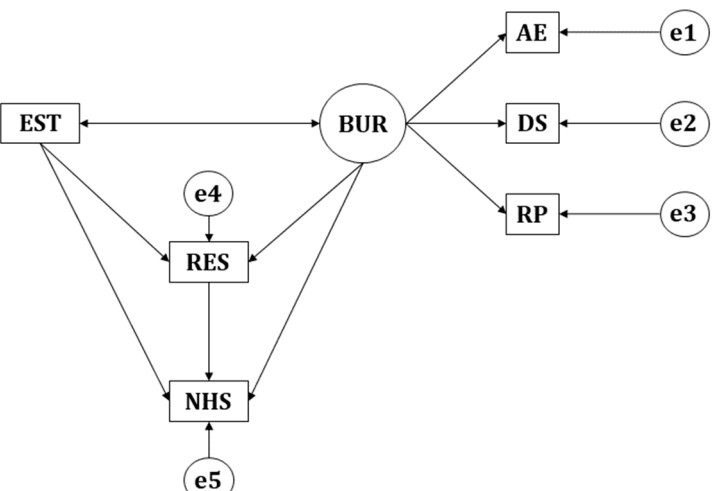

**Figure 1.** Proposed Theoretical Model. **Note:** Stress (EST); Resilience (RES); Burnout Syndrome (BUR); Emotional Exhaustion (EE); Depersonalisation (DS); Number of Study Hours (NHS); Personal Accomplishment (PR).

For the evaluation of the model, the recommendations of Kyriazos (2018) and Maydeu-Olivares (2017) were followed. In this case, goodness-of-fit must be assessed with the Chi-square test, where non-significant *p*-values denote a good fit of the proposed model. Other values such as the comparative fit index (CFI), goodness of fit index (GFI) and incremental reliability index (IFI) were also taken into account, where values higher than 0.900 denote a good fit; however, for the root mean square approximation (RMSEA), values should be lower than 0.100.

### 3. Results

In order to answer hypothesis one, the following analysis has been proposed (Table 2 shows the comparative analysis): For emotional exhaustion, it is evidenced that participants showing high resilience show lower levels (M = 33.92) than those showing medium (M = 38.66) or low resilience (M = 41.80). Likewise, the same is observed for depersonalisation, with lower levels obtained by subjects who show high resilience (M = 14.98), compared to those who show medium and low levels of resilience (M = 16.97; M = 19.72). On the contrary, for personal fulfilment, it is observed that participants showing high levels of resilience (M = 29.66) perform better than those showing medium or low levels (M = 22.48; M = 10.80). Likewise, it is observed that for stress, high levels of resilience (M = 29.25) act positively on this variable, as participants showing medium and low levels (M = 37.67; M = 47.30) obtain higher scores. Finally, for hours of study, it is obtained that participants who show high resilience (M = 5.32) obtain a higher score than those who show a medium or low level (M = 4.85; M = 4.26).

**Table 2.** Comparative study of the total population.

|  |  | *N* | *M* | *DT* | *F* | *P* | *ES (d)* | *95% CI* |
|---|---|---|---|---|---|---|---|---|
| **EA** | **Low Resilience** | 30 | 41.80 | 6.51 | 141.697 | ≤0.05 [a]<br>≤0.05 [b] | 0.776 [a]<br>0.603 [b] | [0.706; 1.439] [a]<br>[0.530; 0.665] [b] |
|  | **Medium Resilience** | 3086 | 38.66 | 7.37 |  |  |  |  |
|  | **High Resilience** | 1001 | 33.92 | 9.23 |  |  |  |  |
| **DS** | **Low Resilience** | 30 | 19.72 | 5.64 | 40.676 | ≤0.05 [a]<br>≤0.05 [b] | 0.700 [a]<br>0.311 [b] | [0.336; 1.065] [a]<br>[0.240; 0.383] [b] |
|  | **Medium Resilience** | 3086 | 16.97 | 6.25 |  |  |  |  |
|  | **High Resilience** | 1001 | 14.98 | 6.80 |  |  |  |  |
| **RP** | **Low Resilience** | 30 | 10.80 | 5.62 | 410.396 | ≤0.05 [a]<br>≤0.05 [b]<br>≤0.05 [c] | 0.960 [a]<br>0.977 [b]<br>0.873 [c] | [2.093; 2.850] [a]<br>[0.902; 1.051] [b]<br>[1.254; 1.978] [c] |
|  | **Medium Resilience** | 3086 | 22.48 | 7.24 |  |  |  |  |
|  | **High Resilience** | 1001 | 29.66 | 7.68 |  |  |  |  |
| **EST** | **Low Resilience** | 30 | 47.30 | 5.59 | 455.941 | ≤0.05 [a]<br>≤0.05 [b]<br>≤0.05 [c] | 0.911 [a]<br>0.773 [b]<br>0.811 [c] | [1.535; 2.280] [a]<br>[0.984; 1.134] [b]<br>[0.950; 1.672] [c] |
|  | **Medium Resilience** | 3086 | 37.67 | 7.36 |  |  |  |  |
|  | **High Resilience** | 1001 | 29.25 | 9.55 |  |  |  |  |
| **NHS** | **Low Resilience** | 30 | 4.26 | 1.89 | 16.298 | ≤0.05 [a]<br>≤0.05 [b] | 0.442 [a]<br>0.270 [b] | [0.078; 0.806] [a]<br>[0.09; 0.630] [b] |
|  | **Medium Resilience** | 3086 | 4.85 | 2.30 |  |  |  |  |
|  | **High Resilience** | 1001 | 5.32 | 2.48 |  |  |  |  |

Note 1: [a] Differences between low resilience and high resilience; [b] Differences between medium resilience and high resilience; [c] Differences between medium resilience and low resilience. **Note 2:** Emotional Exhaustion (EA); Depersonalisation (DS); Personal Accomplishment (PR); Stress (EST); Number of Study Hours (NHS).

To answer hypothesis number three, an analysis based on Pearson's bivariate correlations was developed. Table 3 shows the correlational analysis of the different variables that make up the present study. It is observed that the number of study hours (NHS) is negatively related to stress (r = −0.023) and emotional exhaustion (r = −0.21); however, positive correlations are observed with depersonalisation (r = 0.016), personal fulfilment ($p \leq 0.01$; r = 0.170) and resilience ($p \leq 0.01$; r = 0.102). Regarding stress, it is observed that this variable correlates positively with emotional exhaustion ($p \leq 0.01$; r = 0.534) and depersonalisation ($p \leq 0.01$; r = 0.358); however, negative relationships are shown with personal fulfilment ($p \leq 0.01$; r = −0.553) and resilience ($p \leq 0.01$; r = −0.522). Focusing attention on emotional exhaustion, a positive link is observed with depersonalisation ($p \leq 0.01$; r = 0.481), manifesting negative relationships with personal fulfilment ($p \leq 0.01$; r = −0.381) and resilience (RES) ($p \leq 0.01$; r = −0.522). Likewise, depersonalisation correlates negatively with personal fulfilment (PR) ($p \leq 0.01$; r = −0.230) and resilience ($p \leq 0.01$;

r = 0.176). Finally, it is observed that the variable personal fulfilment obtains a positive relationship with resilience ($p \leq 0.01$; r = 0.553).

**Table 3.** Correlational study of variables.

|  |  | *NHS* | *EST* | *AE* | *DP* | *RP* | *RES* |
|---|---|---|---|---|---|---|---|
| **NHS** | **Pearson correlation** | 1 |  |  |  |  |  |
| **EST** | **Pearson correlation** | −0.023 | 1 |  |  |  |  |
| **AE** | **Pearson correlation** | −0.21 | 0.534 ** | 1 |  |  |  |
| **DP** | **Pearson correlation** | 0.016 | 0.358 ** | 0.481 ** | 1 |  |  |
| **RP** | **Pearson correlation** | 0.170 ** | −0.553 ** | −0.381 ** | −0.230 ** | 1 |  |
| **RES** | **Pearson correlation** | 0.102 ** | −0.522 ** | −0.294 ** | −0.176 ** | 0.553 ** | 1 |

**Note:** ** The correlation is significant at the 0.01 level (bilateral).

Based on the structural equation model developed, the model showed adequate values. Chi-Square analysis yielded a non-significant *p*-value ($X^2$ = 75.450; df = 16; pl = 0.000). Other adjustment indices have been used due to the size and susceptibility of the sample (Tenenbaum and Eklund 2007). In this case, the comparative fit index (CFI), goodness of fit index (GFI) and incremental reliability index (IFI) obtained a value of 0.965, 0.961 and 0.965 respectively. The normalised fit index (NFI) analysis obtained a value of 0.964, the incremental fit index (IFI) was 0.925 and the Tucker-Lewis index (TLI) evidenced a value of 0.963. Finally, the root mean square approximation value (RMSEA) scored 0.051.

Table 4 correspond to research hypotheses number two and number four. In this case, stress has a positive effect on resilience (r = 0.014), but a negative relationship is observed with burnout syndrome ($p \leq 0.001$; r = −0.651). Considering the number of study hours, a positive effect of resilience ($p \leq 0.001$; r = 0.021) and stress ($p \leq 0.001$; r = 0.282) was observed; however, a negative effect of burnout syndrome ($p \leq 0.001$; r = −0.357) on this variable was obtained. Finally, a positive relationship was observed between stress and Burnout syndrome ($p \leq 0.001$; r = 0.824).

**Table 4.** Results of the structural model.

| Variables Effect | Regression Weights | | | | Standardised Regression Weights |
|---|---|---|---|---|---|
|  | Estimations | Estimation Error | Critical Ratio | P | Estimations |
| RES ← EST | 0.001 | 0.003 | 0.320 | 0.749 | 0.014 |
| RES ← BUR | −0.077 | 0.006 | −12.641 | ≤0.001 | −0.651 |
| AE ← BUR | 1.000 |  |  |  | 0.612 |
| DS ← BUR | 0.555 | 0.024 | 23.201 | ≤0.001 | 0.428 |
| RP ← BUR | −1.130 | 0.033 | −33.974 | ≤0.001 | −0.699 |
| NHS ← RES | 0.084 | 0.099 | 0.843 | 0.399 | 0.021 |
| NHS ← EST | 0.076 | 0.012 | 6.511 | ≤0.001 | 0.282 |
| NHS ← BUR | −0.169 | 0.028 | −6.106 | ≤0.001 | −0.357 |
| EST ←→ BUR | 35.697 | 0.137 | 260.868 | ≤0.001 | 0.824 |

## 4. Discussion

The results obtained show that resilience acts as a beneficial element to alleviate the negative effects generated by burnout syndrome and stress. In this case, emotional exhaustion is related to work overload (Jiménez-Ortiz et al. 2019); however, in the academic field it is related to an excessive number of tasks to be performed (González-Valero et al.

2021). Ye et al. (2021) found that a high number of academic tasks act as a stressor for students, negatively influencing their prospects of success in different tasks and leading to a tendency to drop out, and under-preparedness when facing a given academic test. Likewise, depersonalisation consists of the appearance of physical and mental fatigue related to the preparation for a specific academic test (Özhan 2021), with stress playing a fundamental role in the appearance of these symptoms (Ruiz-Calzado and Llorent 2018). In this case, the detrimental effects generated are due to the worry caused by the different tasks, or by the person's perception of his or her work when preparing for, or coping with, a stressful event (Janatolmakan et al. 2021). With regard to personal fulfilment, it is observed that resilience acts positively on this variable. Román-Mata et al. (2020) state that resilience acts as a capacity to adapt and overcome stressful events, which is positively related to achieving success in the academic task and psychological well-being during the preparation of a given task.

Looking at the levels of stress and resilience, it is observed that the latter variable acts positively on stress levels. The research carried out by González-Valero et al. (2022b) establishes that continuous exposure to stress affects the psychological and physical health of individuals, leading to poor performance, low productivity and low motivation when tackling a given task. Rodríguez-Fernández et al. (2018) affirm that resilience acts as an element that favours positive states, through which different events that may be harmful to people's physical and mental health can be successfully faced. This process is mainly due to a dual perspective, where neuroscience and biology act on the brain, the latter being in charge of regulating the cognitive, neurobiological and psychological mechanisms of the person to cope with a response to such stressful events (Vicente de Vera-García and Gabari-Gambarte 2019).

It is observed that resilience acts beneficially on the number of hours of study. León-Hernández et al. (2019) established that being resilient helps to face and develop adequate motivation when facing a certain activity. Despite the fact that psychological, physical and emotional fatigue is generated when facing an academic task, resilience acts beneficially on this task as it helps the development of a positive mindset (Yang and Wang 2022). Furthermore, Suárez-Martel and Martín-Santana (2019) state that emotional intelligence helps to prevent the appearance of disruptive states such as stress and anxiety when preparing for and taking an assessment test.

Focusing on structural equation modelling, it shows positive relationships between stress and resilience. Distant results were obtained by Chakradhar et al. (2022). Similar results to those of this research were found by Baumgartner and Schneider (2021), stating that when stress levels start to increase, symptomatology also increases, necessitating higher emotional competences and resilience. A positive relationship is also observed between the number of hours of study and stress. In view of these findings, Wang et al. (2022) state that the preparation for an evaluative test causes an increase in stress levels in people, as test-takers expect to perform as well as possible and spend as much time preparing for the test as possible. A positive relationship is also shown between burnout syndrome and stress, with Giorgi et al. (2020) stating that continuous exposure to prolonged levels of disruptive states, such as stress and anxiety, leads to emotional exhaustion, which has a negative effect on the preparation for a task, on the motivation developed towards the task, and on the person's own ability to cope with it.

With regard to burnout syndrome, negative relationships are observed with the number of hours of study and resilience. Smith and Emerson (2021) state that burnout is detrimental to the academic environment, negatively affecting the preparation for different tests and tasks. A key element that helps to reduce the effects of this syndrome is the regular practice of physical activity since, during this activity, neurotransmitters are secreted which help to alleviate the effects of burnout (Puertas-Molero et al. 2022; Tong et al. 2021). Furthermore, resilience acts as a key element in the prevention of burnout syndrome symptoms, helping to maintain motivation and a positive attitude towards the performance of any academic activity (Valdivieso-León et al. 2020).

Although this research has fulfilled the objectives of the study, it has a number of limitations, which are listed below. The first of these is related to the nature of the study since, as it is a cross-sectional study, it only offers the relationships of the variables at that point in time. In addition, the instruments used have been validated and have an intrinsic measurement error.

With regard to future perspectives, it is proposed to continue research on this sample, trying to study the effects of the variables over a longitudinal period, in such a way that cause-effect relationships are obtained over time.

## 5. Conclusions

The comparative analysis shows that resilience is a key factor in mitigating the effects of burnout syndrome and stress. It is observed that the higher the levels of resilience, the lower the levels of emotional exhaustion, depersonalisation and stress, and the higher the levels of personal fulfilment, as well as the higher the number of hours of study.

The structural equation model shows a positive association between burnout syndrome and stress, with a positive correlation between stress and the number of hours of study. In contrast, negative relationships are observed between resilience and burnout syndrome, with another negative relationship between burnout and the number of study hours. In addition, a positive relationship was observed between stress and burnout syndrome. Finally, a positive relationship was observed between the number of study hours and resilience.

**Author Contributions:** Conceptualization, E.M.-I. and M.R.S.-R.; methodology, G.G.-V.; software, P.P.-M.; validation, F.Z.-O., J.M.A.-V. and E.M.-I.; formal analysis, E.M.-I.; investigation, G.G.-V. and P.P.-M.; resources, G.G.-V.; data curation, P.P.-M.; writing—original draft preparation, F.Z.-O.; writing—review and editing, M.R.S.-R.; visualization, P.P.-M.; supervision, F.Z.-O.; project administration, E.M.-I.; funding acquisition, G.G.-V. All authors have read and agreed to the published version of the manuscript.

**Funding:** This research received no external funding.

**Informed Consent Statement:** Informed consent was obtained from all subjects involved in the study.

**Data Availability Statement:** The data used to support the findings of current study are available from the corresponding author upon request.

**Conflicts of Interest:** The authors declare no conflict of interest.

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
