# Peer review of "Burnout Syndrome, Stress and Study Hours in the Selection Process for Educational Teaching Staff: The Role of Resilience—An Explanatory Model"

_socsci, doi:10.3390/socsci12040242_

Round 1
Reviewer 1 Report
The submitted article is loically structured. The research methodology is clear. I appreciate the size of the research sample and the statistical processing of the data (especially the graphical model). I have a small remark about the introduction, where more relevant research could be mentioned.
Author Response
REVIEWER 1
Comment 1
The submitted article is loically structured. The research methodology is clear. I appreciate the size of the research sample and the statistical processing of the data (especially the graphical model). I have a small remark about the introduction, where more relevant research could be mentioned.
Response 1
Thank you very much for your suggestion.
New thematically related research has been added.
Reviewer 2 Report
Dear Author,
The article is original.
The object of the article is well delimited.
The subject is relevant.
The methodology is adequate.
The theoretical approach is actual and relevant.
The conclusions follow from the empirical research and the discussions.
There are punctuation mistakes (e.g.: “are applying. (Real Decreto 276/2007),” ; “the selection board. (Real Decreto 276/2007).” ; “(Freudenberger 1989)”
It is necessary to adapt the references to the norms of the Social Sciences.
Sincerely,
Author Response
REVIEWER 2
Comment 1
Dear Author,
The article is original. The object of the article is well delimited. The subject is relevant. The methodology is adequate. The theoretical approach is actual and relevant. The conclusions follow from the empirical research and the discussions.
Response 1
Thank you very much for your suggestion.
Comment 2
There are punctuation mistakes (e.g.: “are applying. (Real Decreto 276/2007),” ; “the selection board. (Real Decreto 276/2007).” ; “(Freudenberger 1989)”
It is necessary to adapt the references to the norms of the Social Sciences.
Response 2
Thank you very much for your comment.
The wording has been revised and the bibliographical references have been reworded.
Reviewer 3 Report
The study had a good sample size of 4117 and the topic of Burnout Syndrome, Stress and Study Hours is a relevant area of study. Clear description of methodology and detailed results. How about relating the results to research hypotheses proposed on p. 2? There is the need to move the studies mentioned in Discussion section from pages 7- 9 to a section on literature review before discussing the findings with reference to the studies.
Rephrase - being these states due to the stress generated by reaching the longed-for posi-tion, risking this position through an evaluative test (Arias & Amate-Romera, 2019).
Rephrase for clarity of meaning -sometimes the levels do not even rise (Jordan et al., 2020)
p. 2
rephrase - of an academic year, concluding that stress levels
p. 8
Link between these two ideas - Focusing on the structural equation model, it shows positive relationships between stress and resilience. Distant results were obtained by Chakradhar et al. (2022), however, Baumgartner and Scheneider (2021) state that when stress levels begin to increase, symp-tomatology also increases, requiring greater emotional competencies along with resilience?
p. 9
This needs to be elaborated on - In addition, the instruments used, although validated and adapted to the study population, are inherently flawed.
Edit punctuation e.g.,
· tion board. (Real Decreto 276/2007).
· Figure 1). the model
Author Response
REVIEWER 3
Comment 1
The study had a good sample size of 4117 and the topic of Burnout Syndrome, Stress and Study Hours is a relevant area of study. Clear description of methodology and detailed results. How about relating the results to research hypotheses proposed on p. 2? There is the need to move the studies mentioned in Discussion section from pages 7- 9 to a section on literature review before discussing the findings with reference to the studies.
Response 1
Thank you very much for your comment.
The data analyses have been linked to the research hypotheses. In addition, we have proceeded to carry out what the literature review indicates
Comment 2
Rephrase - being these states due to the stress generated by reaching the longed-for posi-tion, risking this position through an evaluative test (Arias & Amate-Romera, 2019).
Rephrase for clarity of meaning -sometimes the levels do not even rise (Jordan et al., 2020) p. 2
rephrase - of an academic year, concluding that stress levels p. 8
Response 2
Thank you very much for your suggestions for improvement.
The two sentences in question have been reworded.
Comment 3
Link between these two ideas - Focusing on the structural equation model, it shows positive relationships between stress and resilience. Distant results were obtained by Chakradhar et al. (2022), however, Baumgartner and Scheneider (2021) state that when stress levels begin to increase, symp-tomatology also increases, requiring greater emotional competencies along with resilience? P.9
Response 3
Thank you very much for your suggestions for improvement.
The two ideas have been brought together
Comment 4
This needs to be elaborated on - In addition, the instruments used, although validated and adapted to the study population, are inherently flawed.
Edit punctuation e.g.,
- tion board. (Real Decreto 276/2007).
- Figure 1). the model
Response 4
Thank you very much for your suggestions for improvement.
The two ideas have been brought together
Round 2
Reviewer 3 Report
Thank you for your revision. There is still the need to move the studies mentioned in Discussion section from pages 7- 9 to a section on literature review before discussing the findings with reference to the studies.
Author Response
Comment 1
Thank you for your revision. There is still the need to move the studies mentioned in Discussion section from pages 7- 9 to a section on literature review before discussing the findings with reference to the studies.
Response 1
Thank you very much for your comment. Your comments have been added.